# TLR2 and TLR4 Modulate Mouse Ileal Motility by the Interaction with Muscarinic and Nicotinic Receptors

**DOI:** 10.3390/cells11111791

**Published:** 2022-05-30

**Authors:** Elena Layunta, Raquel Forcén, Laura Grasa

**Affiliations:** 1Department of Medical Biochemistry and Cell Biology, Institute of Biomedicine, University of Gothenburg, Medicinaregatan 9C, 41390 Gothenburg, Sweden; elena.layunta@medkem.gu.se; 2Departamento de Farmacología, Fisiología y Medicina Legal y Forense, Facultad de Veterinaria, Universidad de Zaragoza, 50013 Zaragoza, Spain; r.forcen.90@gmail.com; 3Instituto de Investigación Sanitaria de Aragón (IIS Aragón), 50009 Zaragoza, Spain; 4Instituto Agroalimentario de Aragón—IA2—(Universidad de Zaragoza-CITA), 50013 Zaragoza, Spain

**Keywords:** microbiota, toll-like receptors, intestinal motility, nicotinic receptors, muscarinic receptors

## Abstract

Irritable bowel syndrome (IBS) is a chronic functional bowel disorder characterized by intestinal dysmotility. Changes in intestinal microbiota (dysbiosis) can lead to alterations in neuro-muscular functions in the gut. Toll-like receptors (TLRs) 2 and 4 recognize intestinal bacteria and are involved in the motor response induced by gastrointestinal (GI) neurotransmitters. Acetylcholine (ACh) is a well-known neurotransmitter involved in the regulation of GI motility. This study aimed to evaluate the role of TLR2 and TLR4 in the intestinal motor-response induced by ACh in the mouse ileum, as well as the expression and function of the muscarinic and nicotinic ACh receptors. Muscle contractility studies showed that the contractions induced by ACh were significantly lower in TLR2^−/−^ and TLR4^−/−^ with respect to WT mice. In WT mice, the contractions induced by ACh were reduced in the presence of AF-DX AF-DX 116 (a muscarinic ACh receptor (mAChR) M2 antagonist), 4-DAMP (a mAChR M3 antagonist), mecamylamine (a nicotinic AChR receptor (nAChR) α3β4 antagonist) and α-bungarotoxin (a nAChR α7 antagonist). In TLR2^−/−^ mice, the contractions induced by ACh were increased by AF-DX 116 and mecamylamine. In TLR4^−/−^ mice, the contractions induced by ACh were reduced by α-bungarotoxin and 4-DAMP. The mRNA and protein expressions of M3 and α3 receptors were diminished in the ileum from TLR2^−/−^ and TLR4^−/−^ with respect to WT mice. However, the levels of mRNA and protein of β4 were diminished only in TLR4^−/−^ but not in TLR2^−/−^ mice. In conclusion, our results show that TLR2 and TLR4 modulates the motor responses to ACh in the mouse ileum. TLR2 acts on muscarinic M2 and M3 and nicotinic α3β4 ACh receptors, while TLR4 acts on muscarinic M3 and nicotinic α3β4 and α7 ACh receptors.

## 1. Introduction

Irritable bowel syndrome (IBS) is a chronic functional bowel disorder characterized by altered visceral sensitivity, functional brain alterations, secretory dysfunctions and intestinal dysmotility. Although the etiology of IBS has not yet been completely elucidated, previous studies have indicated that increased epithelial permeability, microbial gut alteration called dysbiosis, modified expression of immune mediators, visceral hypersensitivity and a dysfunctional brain–gut axis could be important factors in the cause of IBS [1].

The current hypothesis explaining the intestinal motility disturbances observed in patients with IBS is that dysbiosis-driven mucosal alterations induce the production of several inflammatory/immune mediators, which affect neuro-muscular functions in the gut [2]. Additionally, growing evidence suggests that microbiota can directly affect enteric nerves and smooth muscle cell functions through its metabolic products or bacterial molecular components translocated from the intestinal lumen, affecting intestinal motility [2,3]. In fact, observations in germ-free animals suggest that microbiota modulates the expression of genes involved in the functional responses of the motor apparatus [4]. Novel therapeutics in IBS include the administration of probiotics that improve the clinic of IBS [5,6]. Indeed, previous *in vitro* studies show that some bacteria, such as *Escherichia coli Nissle 1917* and *Lactobacillus rhamnosus GG*, regulate the contractility of the human colonic smooth muscle [7,8].

Innate immune Toll-like receptors (TLRs) recognize bacteria by detecting different molecular-associated molecular patterns (MAMPs) [9,10]. Among TLR, the most important bacteria-sensing receptors in the gut are TLR2 and TLR4, which play crucial roles in the innate immune system [10]. TLR2, TLR4 and NFκB are expressed on the intestinal smooth muscle, which can be induced to generate inflammatory mediators as well as reactive oxygen species (ROS) that are proposed to contribute to reduced smooth muscle contractility [11]. In addition to the smooth muscle, the expression of TLR2 has been found in enteric neurons and the glia of the mouse ileum [12] and TLR4 has been localized in enteric neurons of mouse, rat and human intestines [13,14], emerging as potential mediators between microbiota and the enteric neuromuscular apparatus. Indeed, TLR2 and TLR4 would be involved in the motor response induced by some gastrointestinal (GI) neurotransmitters like nitric oxide (NO), serotonin (5-HT) or hydrogen sulfide (H_2_S) [15,16,17,18]. However, the role of TLR2 and TLR4 in contractile response induced by the neurotransmitter acetylcholine (ACh) is largely unknown.

ACh released from parasympathetic nerves plays an important role in the regulation of GI motility. Muscarinic receptors on enteric neurons and muscle cells are targets of ACh. Molecular studies have demonstrated the expression of five muscarinic receptors (M1-M5) in the GI tract, which are coupled to membrane-associated GTP-binding proteins (G-proteins) [19]. M2 and M3 are the main receptor subtypes expressed on muscle cells and mediate contraction induced by ACh [19,20]. The M2 receptors are also expressed on enteric cholinergic nerves and regulate ACh release [21]. Functional studies using M2 or M3 receptor knockout (KO) mice have indicated that the M2 and M3 receptors cause ileum contraction through different mechanisms, but in wild-type mice, a synergistic pathway requiring both subtypes is activated [20].

On the other hand, in the nervous system there are also the nicotinic acetylcholine receptors (nAChRs), which are ligand-gated cation channels composed of pentameric combinations of 11 subunits (α2–α9; β2–β4) [22]. Immunohistochemical studies have revealed that guinea pig and mouse myenteric neurons express nAChRs composed of α3, α5, α4, α7, β2 and β4 [23,24].

While it has been widely reported that ACh and microbiota modulate the motor response in the GI tract, there is a knowledge gap regarding the involvement of TLR2 and TLR4 in the intestinal contractile activity mediated by ACh. Therefore, this study aimed to evaluate the role of TLR2 and TLR4 in the intestinal motor-response induced by ACh in mice, as well as the expression and function of the muscarinic and nicotinic ACh receptors.

## 2. Materials and Methods

### 2.1. Animals

The Ethics Committee for Animal Experiments from University of Zaragoza approved all the experiments developed in this study (Project License PI03/16) based on the Spanish Policy for Animal Protection RD53/2013, in agreement with the European Union Directive 2010/63. Free-pathogen animals were maintained in a 12-h light/dark cycle with water and food ad libitum. C57BL/10 mouse strains knockout for TLR2 (TLR2^−/−^) and for TLR4 (TLR4^−/−^) were acquired from Ignacio Aguiló and bred at the Centro de Investigación Biomédica de Aragón (CIBA), Zaragoza, Spain [25]. In this study, TLR2^−/−^ and TLR4^−/−^ knockout (KO) males, 8 to 12 weeks old and age-matched with wild-type (WT) animals, were used.

### 2.2. Muscle Contractility Studies

Animals were sacrificed by cervical dislocation and the ileum was rapidly harvested and located in an ice-cold Krebs buffer. Ileum segments (10 mm-length) with intact mucosa were suspended in the longitudinal direction of the smooth muscle fibers in an organ bath, thermostatically controlled at 37 °C with Krebs solution and gassed continuously (95% O_2_ and 5% CO_2_). Each segment was connected to an isometric force transducer (Pioden UF1, Graham Bell House, Canterbury, UK) and stretched passively to an initial tension of 0.5 g. The mechanical activity signal was amplified (The Mac Lab Bridge Amp, AD Instruments Inc., Milford, MA, USA) to a range of 2 mV and recorded to be analyzed and digitized (two samples/second/channel) in the Mac Lab System/8e computer program (AD Instruments Inc., Milford, MA, USA). Before the experiments, the segments were equilibrated for 1 h in Krebs buffer, changed every 20 min. After this period of equilibration, the spontaneous motility patterns of ileum from WT and TLRs KO animals were acquired. Then, acetylcholine (ACh), in a 0.1–10 µM concentration range, was added to the organ bath accumulatively, increasing concentrations every 20 min.

The ACh concentration–response curves in the WT ileum were compared with the ones acquired from TLRs^−/−^ KO animals. To address the role of the muscarinic and nicotinic ACh receptors on the responses triggered by ACh, the ileum sections were incubated for 15 min with 0.2 µM AF-DX 116, a selective muscarinic acetylcholine receptor (mAChR) M2 antagonist; 0.1 µM 4-DAMP, a specific mAChR M3 antagonist; 10 µM mecamylamine, non-selective nicotinic acetylcholine receptors (nAChR) α3 and β4 antagonists; or 2 nM α-bungarotoxin, a specific α7 nicotinic acetylcholine receptor (nAChR) antagonist, prior to the performance of the 0.1–10 µM ACh concentration–response curves. Previous studies have reported that similar concentration ranges of these antagonists induce effects in the neurotransmission of the intestinal smooth muscle [24,26,27].

All intestinal sections in the study presented spontaneous contractions. Then, maximum contraction amplitudes triggered by ACh with or without antagonist were measured. The maximum contraction was calculated as the difference between the maximum and the minimum (Max–Min) and expressed as a percentage of increase with respect to control spontaneous motility. These data were used to calculate concentration–response curves using non-linear regression. EC_50_ values (the concentration of a drug that gives half-maximal response) were estimated in the absence or presence of the antagonists.

### 2.3. Gene Expression by Real-Time PCR

The relative gene expression of muscarinic (M2 and M3) and nicotinic (α3, β4 and α7) ACh receptors in the ileum from WT and TLRs^−/−^ KO animals was studied by real-time PCR. RNA extractions were carried out with the RNeasy mini kit (Qiagen, Hilden, Germany) and then used as a template to synthesize cDNA by using the NX M-MuLV Reverse Transcriptase kit (Lucigen, Middleton, WI, USA). The obtained cDNA was used to measure the transcriptomic levels by SYBR Green and specific primers indicated in Table 1. RT-PCR was run using the StepOne Plus Real-Time PCR System (Life Technologies, Carlsbad, CA, USA). The reaction mixture (10 µL) was composed of FastStart Universal SYBR Green Master (Roche, Mannheim, Germany), 0.5 µL of each primer 30 µM, 2.5 µL of sterile water and 2 µL of cDNA template (200 ng). Each biological sample was run in triplicate, and the C_t_ mean was calculated. Relative mAChR or nAChR mRNA levels in each group of animals (WT, TLR2^−/−^ or TLR4^−/−^) was expressed as ΔC_t_ = C_t_ receptor − C_t_ calibrator. GAPDH and HPRT housekeeping genes were used as calibrators after the verification of their stability in our experiment. Then, the relative receptor mRNA expression was calculated as ΔΔC_t_ = ΔC_t_ TLR^−/−^ − ΔC_t_ WT. Finally, the relative mRNA expression was expressed as fold change (=2^−ΔΔCt^).

### 2.4. Western Blotting

Mouse ileum segments were homogenized in a Polytron homogenizer (DI 25 Basic, IKA-WERKE, Germany) in cold-RIPA lysis solution with 100 mM PMSF, 0.3% aprotinin and 0.1% sodium orthovanadate. The homogenates were centrifuged at 4 °C during 20 min at 15,000× *g*. The supernatant with the proteins was kept at –80 °C until use. The protein concentration was measured by the Bradford method (Bio-Rad, Hercules, CA, USA). Samples were solubilized with 2 × SDS sample solution containing pH 6.8 4 × Tris·Cl/SDS (0.5 M Tris and 0.4% SDS), 20% glycerol, 4% SDS, 0.2 M DTT and 0.001% bromophenol blue and later incubated at 100 °C for 5 min.

The same amount of ileum protein (60 µg) from WT, TLR2^−/−^ and TLR4^−/−^ animals were loaded and separated by using electrophoresis on 10% sodium dodecyl sulfate-polyacrilamide gel. After electrophoresis, proteins were transferred to a PVDF transfer membrane (Immobilon, Millipore, Bedford, MA, USA). Then, membranes were blocked in a 0.05% Tween 20/PBS (PBST) solution with 4% non-fat dried milk and 0.01% BSA for 1 h at RT. Membranes were incubated overnight at 4 °C with rabbit primary antibodies against M3 muscarinic receptor (1:2000), nicotinic ACh receptor α3 (1:500) or nicotinic ACh receptor β4 (1:200). All the primary antibodies were kindly donated by Alomone Labs (Jerusalem, Israel) and diluted in PBST with 1% non-fat dried milk and 1% BSA. The detection of primary antibodies was carried out by using a peroxidase-conjugated goat anti-rabbit antibody (1:5000, Santa Cruz Biotechnology, Santa Cruz, CA, USA) for 1 h at RT and the Amersham ECL Plus Western Blotting Detection System (GE Healthcare, Chicago, IL, USA). In negative control experiments, secondary antibodies were included and primary antibodies were absent. The protein signal was obtained by VersaDocTM (Imaging System, Bio-Rad). After stripping, membranes were reprobed with an anti-β-actin antibody (1:500, Santa Cruz Biotechnology, Santa Cruz, CA, USA) for the normalization of protein load. The M3, α3 and β4/β-actin protein ratios were measured in densitometry units by using Quantity One 1-D Analyses Software (Bio-Rad), and the data were showed as a percentage (100%) of the control results.

### 2.5. Data Analysis and Statistics

Data were expressed as the mean ± SEM with *n* indicating the number of ileal segments or mice used. The data were calculated by using the software GraphPad Prism version 5.00 (GraphPad Software, San Diego, CA, USA), where the *p*-values < 0.05 were considered statistically significant. Differences in the responses to the different ACh concentrations in WT, TLR2^−/−^ and TLR4^−/−^ animals were analyzed by one-way analysis of variance (one-way ANOVA) followed by Bonferroni’s post hoc test. Differences in the responses to the different concentrations of the drugs in WT, TLR2^−/−^ and TLR4^−/−^ mice were compared by two-way analysis of variance (two-way ANOVA). EC_50_ values were calculated using a conventional concentration–response curve with a variable slope and expressed as LogEC_50_ (95% Confidence Intervals), and concentration (µM). Differences in the mRNA and protein levels in TLR2^−/−^ and TLR4^−/−^, with respect to WT mice, were compared by Mann–Whitney U-test.

### 2.6. Drugs and Solutions

Normal pH 7.4 Krebs buffer was composed of 120 mM NaCl, 4.7 mM KCl, 2.4 mM CaCl_2_, 1.2 mM MgSO_4_, 24.5 mM NaHCO_3_, 1.0 mM KH_2_PO_4_ and 5.6 mM glucose.

4-DAMP, AF-DX 116 and mecamylamine HCl were acquired from Sigma (Madrid, Spain). The α-Bungarotoxin was acquired from Alomone Labs (Jerusalem, Israel).

## 3. Results

### 3.1. Effect of ACh on Ileal Motility of WT, TLR2^−/−^ and TLR4^−/−^ Mice

Concentration–response curves to ACh (0.1–10 µM) were performed in whole strips of mouse ileum suspended in the longitudinal direction in an organ bath. ACh induced a concentration-dependent contraction in the longitudinal smooth muscle of ileum from WT (*p* < 0.001), TLR2^−/−^ (*p* < 0.001) and TLR4^−/−^ (*p* < 0.001) mice (Figure 1). The EC_50_ values of ACh in WT, TLR2^−/−^ and TLR4^−/−^ mice are described in Table 2. The contractions induced by ACh were significantly lower in TLR2^−/−^ (*p* < 0.001) and TLR4^−/−^ (*p* < 0.001) in respect to WT mice (Figure 1).

### 3.2. Effects of Muscarinic and Nicotinic ACh Receptors Antagonists on ACh-Evoked Response in Ileum from WT, TLR2^−/−^ and TLR4^−/−^ Mice

We analyzed the effects of AF-DX 116 (a selective mAChR M2 antagonist, 0.2 µM), 4-DAMP (a selective mAChR M3 antagonist, 0.1 µM), mecamylamine (a non-selective nAChR α3 and β4 receptor antagonist, 10 µM) and α-bungarotoxin (a selective nAChR α7 antagonist, 2 nM) on the contractile response of ACh (0.1–10 µM) in WT, TLR2^−/−^ and TLR4^−/−^ mice (Figure 2). In WT mice, the contractions induced by ACh were reduced in the presence of AF-DX 116 (*p* < 0.001), 4-DAMP (*p* < 0.001), mecamylamine (*p* < 0.001) and α-bungarotoxin (*p* < 0.01) (Figure 2A,B). In TLR2^−/−^ mice, the contractions induced by ACh were increased by AF-DX 116 (*p* < 0.1) (Figure 2C) and mecamylamine (*p* < 0.01) (Figure 2D), but unmodified by 4-DAMP or α-bungarotoxin (Figure 2C,D).

In TLR4^−/−^ mice, the contractions induced by ACh were reduced by α-bungarotoxin (*p* < 0.1) (Figure 2F) and even blocked in the presence of 4-DAMP (*p* < 0.001) (Figure 2E). However, mecamylamine and AF-DX 116 did not modify the ACh contractions in TLR4^−/−^ mice (Figure 2E,F). The EC_50_ values of ACh in the presence of AF-DX 116, 4-DAMP, mecamylamine and α-bungarotoxin in WT, TLR2^−/−^ and TLR4^−/−^ mice are described in Table 2.

### 3.3. mRNA and Protein Expression Levels of Muscarinic and Nicotinic ACh Receptors in WT, TLR2^−/−^ and TLR4^−/−^ Mice

We studied the mRNA expression of muscarinic (M2 and M3) and nicotinic (α3, β4, and α7 subunits) ACh receptors in the ileum from WT, TLR2^−/−^ and TLR4^−/−^ mice to evaluate the role of these receptors in the responses evoked by ACh in the different groups of animals (Figure 3 and Figure 4). The mRNA expression of M3, α3 and β4 receptors was diminished in the ileum from TLR2^−/−^ and TLR4^−/−^ with respect to WT mice Figure 3B, and Figure 4A,C). The mRNA expression of the M2 and α7 receptors was not modified in TLR2^−/−^ and TLR4^−/−^ with respect to WT mice (Figure 3A and Figure 4B).

To check whether the observed decrease in the mRNA expression of the M3, α3 and β4 receptors corresponded to a decrease in the protein levels of these receptors, we carried out Western blotting experiments (Figure 5). The protein levels of M3 and α3 were diminished in TLR2^−/−^ and TLR4^−/−^ with respect to WT mice (Figure 5A,B). However, the protein levels of β4 were diminished only in TLR4^−/−^ but not in TLR2^−/−^ mice (Figure 5C).

## 4. Discussion

TLR2 and TLR4, recognized receptors of Gram-positive and Gram-negative bacteria, contribute to maintaining the normal spontaneous motility in the mouse ileum [12,16,32,33]. In this work, we analyzed the influence of TLR2 and TLR4 on the contractile response evoked by ACh in mouse ileum. Our results show that the contractions induced by ACh were significantly lower in TLR2^−/−^ and TLR4^−/−^ compared with WT mice, indicating that both receptors are necessary to maintain the intestinal cholinergic neurotransmission. Other authors have reported the influence of the TLR2 and TLR4 ligands on the intestinal contractions induced by ACh. The lipopolysaccharide (LPS) from *Shigella flexneri* and *Escherichia coli*, both ligands of TLR4, and Pam2CSK4 and Pam3CSK4, which activate TLR2/6 and TLR1/2 heterodimers, respectively, decreased the ACh-induced contractions in primary cultures of colonic human smooth muscle cells [34]. Additionally, LPS from *Escherichia coli* can also decrease the contractions induced by ACh in the rabbit duodenum [35,36]. These data suggest that the lack of TLR2 or TLR4 as well as over-activation of these TLRs would reduce the intestinal motility mediated by ACh.

The effects of ACh on GI motility are mediated by the activation of muscarinic and nicotinic ACh receptors. Among the muscarinic ACh receptors, M2 and M3 are the main receptor subtypes expressed on mouse intestinal smooth muscle cells that mediate contraction induced by ACh [19,20]. Our results show that AF-DX 116, a selective mAChR M2 antagonist [37], and 4-DAMP, a selective mAChR M3 antagonist [37], reduced the contractions induced by ACh in WT mice, corroborating that M2 and M3 receptors mediate the ACh contractile responses in mouse ileum.

In relation to the nicotinic ACh receptors, it has been reported that guinea pig and mouse myenteric neurons express nAChRs composed of α3, α4, α7, β2 and β4 subunits [23,24] Our results show that mecamylamine, a non-selective nAChR α3 and β4 receptor antagonist [38], and α-bungarotoxin, a selective nAChR α7 antagonist [39], reduced the contractions induced by ACh in WT mice, indicating that α3, β4 and α7 subunits of nicotinic receptors are involved in the ACh contractile responses in mouse ileum. Nicotinic receptors can be located presynaptically or postsynaptically [40,41]. While there is strong evidence for the involvement of α3β4 nAChRs in synaptic transmission, the contributions of α7s remain elusive in the myenteric plexus [23]. In fact, mecamylamine exerts the anti-nicotinic action in guinea pig ileum primarily through the blockade of the nicotinic receptor in intrinsic cholinergic ganglia and through an inhibition of acetylcholine liberation from the intrinsic postganglionic nerves, without affecting the direct muscle response to acetylcholine [42]. Mecamylamine also reduced the electrical field stimulation-induced cholinergic contractions in the fundus, jejunum and colon of mice, suggesting that part of the cholinergic response is due to the activation of preganglionic neurons [43]. Nevertheless, the contribution of α7 to the cholinergic transmission seems to be controversial. Our results are in agreement with the studies by Obaid et al., showing that the α7 blockade results in a decrement in the magnitude of the fast excitatory post-synaptic potentials evoked by electrical stimulation in guinea pig enteric nervous system [41]. Additionally, studies in mouse myenteric neurons suggest the possibility of α7 receptors contributing to synaptic transmission or release of neurotransmitters other than ACh [23]. However, it has been shown that α7 antagonists (including α-bungarotoxin) did not affect ACh-induced responses in isolated guinea pig myenteric neurons in culture [24].

Our results show that the deficiency of TLR2 or TLR4 expression would modify the expression pattern of muscarinic and nicotinic ACh receptors, as well as the extent of the involvement of these receptors in the motor response to ACh. Thus, our data indicate an interaction between TLR2 and TLR4 with the cholinergic neurotransmission. In this context, our results show that the deficiency of TLR2 induces a decrease in the protein expression of M3 and α3 receptors compared with WT mice, while the expression of M2, β4 and α7 was not modified. In the ileum of TLR2^−/−^ mice, the contractions induced by ACh were modified by M2 or α3β4 antagonists but unmodified by M3 or α7 blockers. Therefore, TLR2 may contribute to ACh-induced response by regulating the expression and/or interacting with the function of M2, M3 and α3β4 receptors in mouse ileum. The finding that the response to ACh is increased in the presence of M2 and α3β4 blockers in TLR2^−/−^ mice, without a change in the expression of mRNA and protein of these receptors, is surprising and represents a limitation to our study. However, this result could be explained by the fact that both muscarinic receptors and nicotinic receptors may be subjected to post-translational modifications that could modify the affinity of the receptors for ACh and, therefore, its function. The lack of a TLR2 gene in these animals could have induced these post-translational modifications. In fact, post-translational modifications have been described in G protein–coupled receptors [44] and specifically, an N-glycosylation seems to be key in determining M3 receptor distribution and localization [45]. On the other hand, three posttranslational modifications are known for the nicotinic acetylcholine receptor family: glycosylation, phosphorylation and palmitoylation [46].

In relation to TLR4^−/−^ mice, the levels of protein expression of M3, α3, and β4 receptors were reduced compared with WT mice, while the expression of M2 and α7 was not modified. In the ileum of TLR4^−/−^ mice, the contractions induced by ACh were modified by M3 or α7 antagonists but unmodified by M2 or α3 β4 blockers. Therefore, TLR2 may contribute to the ACh-induced response by regulating the expression and/or interacting with the function of M3, α3β4 and α7 receptors in mouse ileum.

Previous data obtained in our laboratory have already reported that TLRs would control intestinal motility by the sulfide system modulation [18], the serotonergic system [16,17] and by the direct microbiota–TLRs interaction [47]. However, this is the first time that data show how TLRs are involved in intestinal contractility directly through one of the main mediators, ACh, and its specific muscarinic and nicotinic receptors. Previous studies have highlighted that TLR activation on B cells triggers cholinergic activity by the increasing of ACh synthesis. In turn, ACh produced in B cells would downregulate immunity, which suggests a bidirectional regulation [48]. Our results are in agreement with previous studies showing that deficiency of TLR2 results in gut dysmotility [12].

Additionally, contractility studies carried out in coronary arteries from TLR2^−/−^ mice showed that the lack of TLR2 interferes in contractile response and other mediators like nitric oxide synthase (NOS) could be involved [49]. In turn, nicotinic ACh receptor activation would mediate the TLR2 proinflammatory pathways involved in wound repair [50]. Other studies have described that the activation of TLR4 would control the expression of α7 nACh receptors in rodent microglia [51]. Although our results show that the absence of TLR4 alters the gut motility through changes in the expression or function of ACh muscarinic and nicotinic receptors, other non-cholinergic pathways have been reported to be involved [15].

In conclusion, our results show that TLR2 and TLR4 are involved in the motor responses to ACh in the mouse ileum. TLR2 acts on muscarinic M2 and M3 and nicotinic α3β4 ACh receptors, while TLR4 acts on muscarinic M3 and nicotinic α3β4 and α7 ACh receptors.

## Figures and Tables

**Figure 1 cells-11-01791-f001:**
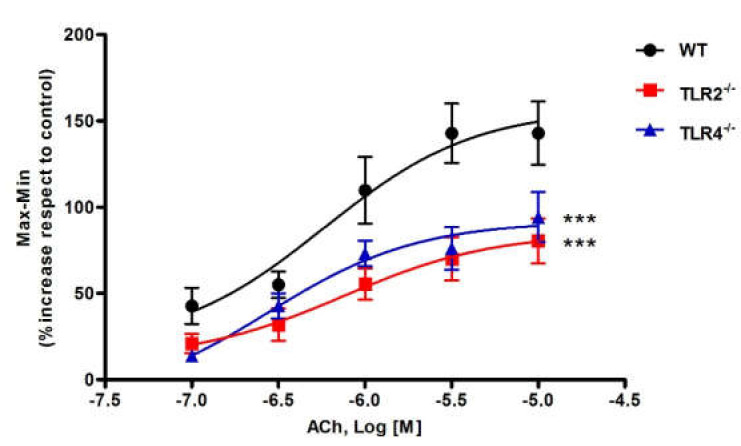
Concentration–response curves to ACh in longitudinal ileum from WT, TLR2^−/−^ and TLR4^−/−^ mice. The results are the mean values of amplitude (Max–Min, expressed as a percentage of increase compared to the control) and the vertical bars indicate SEM (*n* ≥ 16 ileal segments for each group of mice). *** *p* < 0.001 vs. WT mice.

**Figure 2 cells-11-01791-f002:**
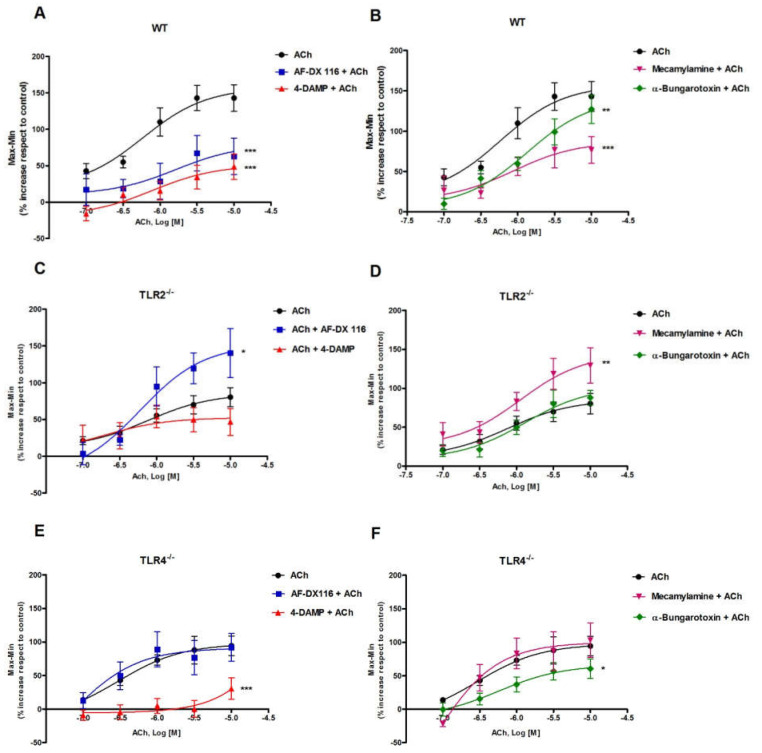
Effect of the incubation for 15 min with AF-DX 116 (0.2 µM), 4-DAMP (0.1 µM), mecamylamine (10 µM) or α-bungarotoxin (2 nM) on the concentration–response curves to ACh (0.1–10 µM) in longitudinal ileum from WT (**A**,**B**), TLR2^−/−^ (**C**,**D**) and TLR4^−/−^ (**E**,**F**) mice. The results are the mean values of amplitude (Max–Min, expressed as the percentage of increase compared to the control) and the vertical bars indicate SEM (*n* ≥ 8 ileal segments for each antagonist from each group of mice). * *p* < 0.05; ** *p* < 0.01; *** *p* < 0.001 vs. ACh.

**Figure 3 cells-11-01791-f003:**
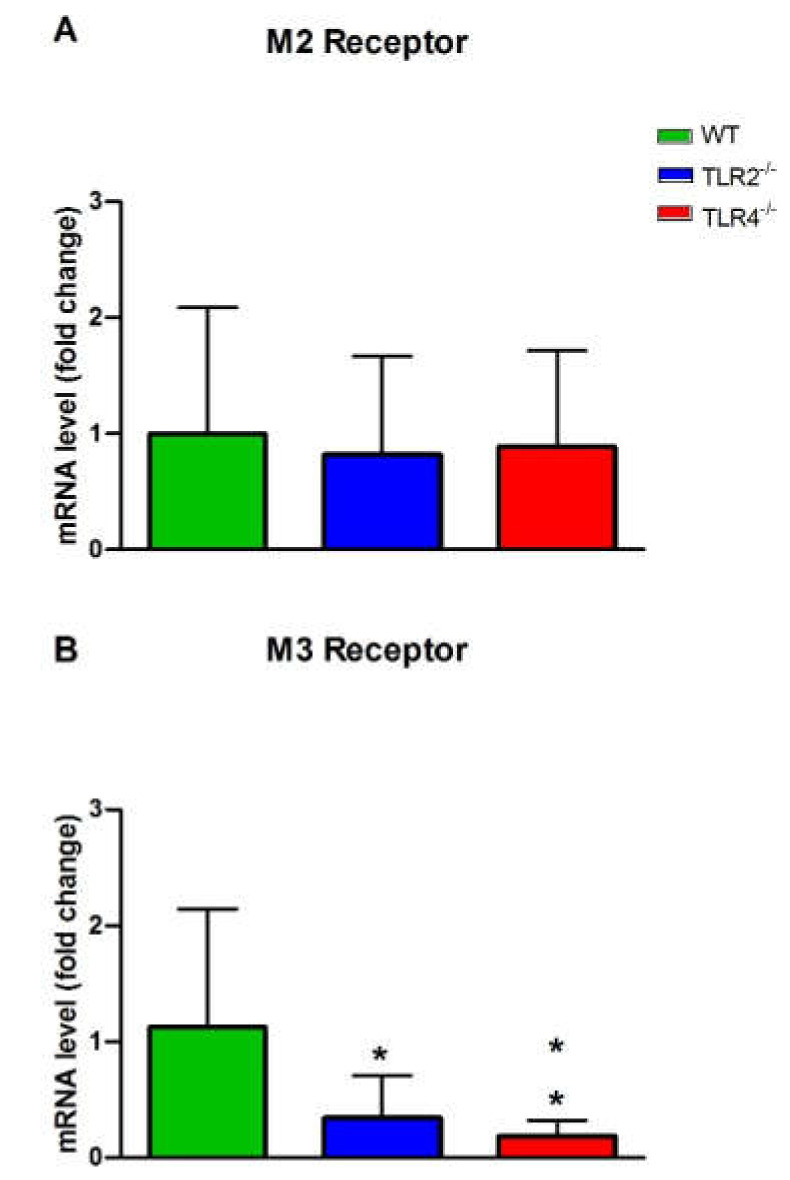
Real time PCR analysis of M2 (**A**) and M3 (**B**) mACh receptors’ mRNA expression in WT, TLR2^−/−^ and TLR4^−/−^ mice ileum. The data are the mean, and the vertical bars indicate SEM of at least five animals per group. * *p* < 0.05; ** *p* < 0.01 vs. WT mice.

**Figure 4 cells-11-01791-f004:**
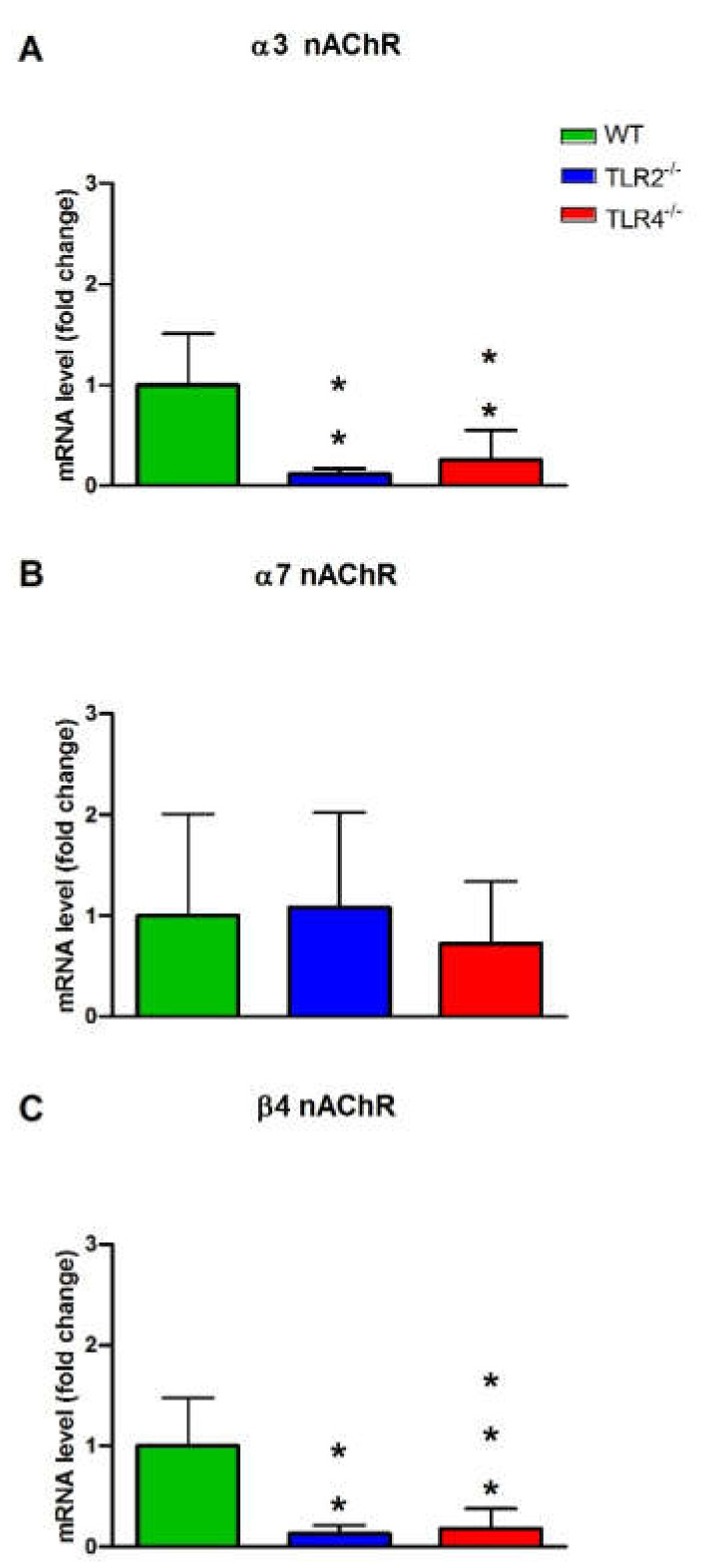
Real time PCR analysis of α3 (**A**), α7 (**B**) and β4 (**C**) nACh receptors’ mRNA expression in WT, TLR2^−/−^ and TLR4^−/−^ mice ileum. The data are the mean, and the vertical bars indicate SEM of at least five animals per group. ** *p* < 0.01; *** *p* < 0.001 vs. WT mice.

**Figure 5 cells-11-01791-f005:**
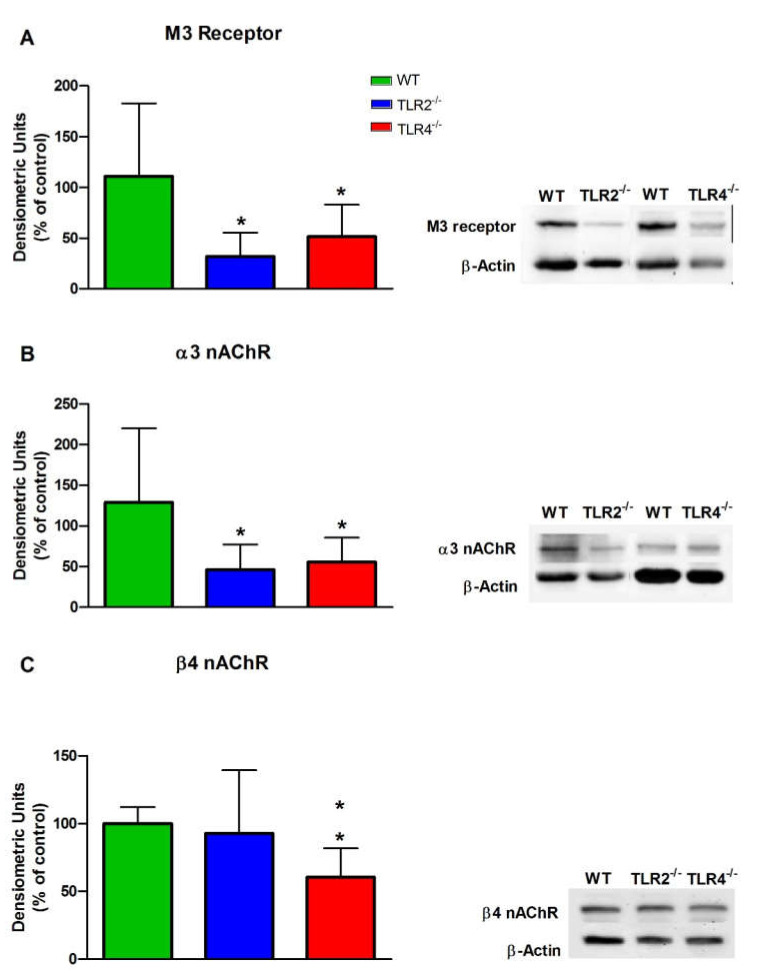
Western blotting of the M3 muscarinic ACh receptor (**A**) and the α3 (**B**) and β4 (**C**) nicotinic ACh receptors in the ileum from WT, TLR2^−/−^ and TLR4^−/−^ mice. The results are the mean values, and the vertical bars indicate SEM of at least five animals per group. * *p* < 0.05; ** *p* < 0.01 vs. WT mice.

**Table 1 cells-11-01791-t001:** Real-time PCR primers used for the mRNA quantification of mAChRs and nAChRs in mouse ileum.

Gene	Reference	GenBank Accession Number	Forward and Reverse Primers
**mAChRs M2**	[28]	NM_203491.3	CGGACCACAAAAATGGCAGGCAT CCATCACCACCAGGCATGTTGTTGT
**mAChRs M3**	[28]	NM_033269.4	CCTCTTGAAGTGCTGCGTTCTGACC TGCCAGGAAGCCAGTCAAGAATGC
**nAChRs α3**	[29]	NM_145129.3	GTGGAGTTCATGCGAGTCCCTG TAAAGATGGCCGGAGGGATCC
**nAChRs α7**	[30]	NM_007390.3	CAGCAGCTATATCCCCAATGG GGCTCTTTGCAGCATTCATAGA
**nAChRs β4**	[31]	NM_148944.4	TGTACAACAATGCCGATGGG CCTGTGGGTTCACTGTCCTT
**HPRT**	[16]	NM_013556.2	CTGGTGAAAAGGACCTCTCGAA CTGAAGTACTCATTATAGTCAAGGGCAT
**GAPDH**	[16]	NM_008084	AACGACCCCTTCATTGAC TCCACGACATACTCAGCAC

**Table 2 cells-11-01791-t002:** Values of LogEC_50_ with 95% Confidence Intervals, EC_50_ (µM) and the number of ileal segments analyzed (*n*) of the concentration–response curves to ACh performed in ileal segments from WT, TLR2^−/−^ and TLR4^−/−^ mice in the presence or absence of different antagonists of muscarinic and nicotinic ACh receptors.

	WT		TLR2^−/−^		TLR4^−/−^	
	LogEC_50_	EC_50_ (µM)	LogEC_50_	EC_50_ (µM)	LogEC_50_	EC_50_ (µM)
	(95% Confidence Intervals)	*n*	(95% Confidence Intervals)	*n*	(95% Confidence Intervals)	*n*
**ACh**	−6.234	0.58	−6.123	0.75	−6.560	0.27
	(−6.944 to −5.524)	*n* = 16	(−6.946 to −5.300)	*n* = 29	(−7.336 to −5.784)	*n* = 25
**AF-DX 116 + ACh**	−5.787	1.63	−6.224	0.59	−7.140	0.07
	(−7.786 to −3.788)	*n* = 13	(−6.963 to −5.484)	*n* = 11	(−10.268 to −4.011)	*n* = 11
**4-DAMP + ACh**	−6.106	0.78	−6.727	0.18	-	-
	(−7.238 to −4.974)	*n* = 9	(−10.217 to −3.237)	*n* = 8		*n* = 12
**Mecamylamine + ACh**	−6.063	0.86	−5.949	1.12	−7.178	0.07
	(−7.330 to −4.797)	*n* = 8	(−6.801 to −5.097)	*n* = 8	(−9.097 to −5.259)	*n* = 11
**α-Bungarotoxin + ACh**	−5.874	1.33	−5.890	1.28	−6.245	0.56
	(−6.385 to −5.362)	*n* = 10	(−6.612 to −5.169)	*n* = 11	(−7.162 to −5.329)	*n* = 9

## Data Availability

The data from this work are available in the article.

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
