# Peer review of "TLR2 and TLR4 Modulate Mouse Ileal Motility by the Interaction with Muscarinic and Nicotinic Receptors"

_cells, 2022, doi:10.3390/cells11111791_

Round 1

Reviewer 1 Report

In the current manuscript “TLR2 and TLR4 modulate mouse ileal motility by the regulation of muscarinic and nicotinic receptors” by  Layunta et al. showed the modulation of motor responses in the mouse ileum by TLR2 and TLR4 signaling. 

Comments

  1. The mRNA expression of M2,M3, α3 nAChR, α7 nAChR and β4 nAChR should be validated after treatment with acetylcholine and there antagonists in the ileal tissues.
  2. Figure 1 and 2, Appropriate statistical test should be performed to assess the significant differences between the groups.

Author Response

Reviewer 1

In the current manuscript “TLR2 and TLR4 modulate mouse ileal motility by the regulation of muscarinic and nicotinic receptors” by Layunta et al. showed the modulation of motor responses in the mouse ileum by TLR2 and TLR4 signaling.

Comments

The mRNA expression of M2,M3, α3 nAChR, α7 nAChR and β4 nAChR should be validated after treatment with acetylcholine and there antagonists in the ileal tissues.

Answer: We appreciate the suggestion of the reviewer. We would like to clarify that the motility experiments have been performed in vitro in an organ bath. Acetylcholine and its antagonists were added to the bath and the effect of these drugs on intestinal smooth muscle was studied. Each ileum segment spent at least 3 hours in the organ bath and was therefore subjected to a situation of stress and tissue damage. For this reason, we believe that the mRNA levels under these conditions would be altered by environmental conditions and would not correspond to reality.

Figure 1 and 2, Appropriate statistical test should be performed to assess the significant differences between the groups.

Answer: Thank you for the suggestion. Statistical analysis are shown in the text of the manuscript (lines 214-217, 224-232, 257-260). However, and for a better understanding of the work, we have added the statistical analysis in Figures 1 and 2.

Reviewer 2 Report

In the manuscript  “TLR2 and TLR4 modulate mouse ileal motility by the regulation of muscarinic and nicotinic receptors”, Layunta et al. aim to demonstrate that TLR2 and TLR4 signaling modulates the motor responses to ACh in the mouse ileum as the deficiency of TLR2 or TLR4 expression affects the expression pattern of muscarinic and nicotinic ACh receptors.

Data interpretation and conclusions:

Overall, the study design and purpose as well as the conclusions need to be clarified and described in more detail. While the authors described the goal of the study to demonstrate that TLR2 regulates muscarinic M2 and M3 and nicotinic α3β4 Ach receptors and TLR4 M3, α3β4 and  α7 Ach receptors, the ‘signaling and regulation’ are not truly defined.

If regulation is protein expression, then the data assessing receptors in the two different mouse models need to be moved up to build the foundation for the subsequent study. However, as described below there are improvements required to show convincing evidence that there is a different in receptor expression upon TLR2 and TLR4 knock-out.

The authors should address if in the absence of knowledge about receptor presence the response to Ach in Figures 1-2 is meaningful.

In terms of signaling, the authors demonstrate a reduced effect of Ach on muscle contractility in both knock-out models compared to wild-type. If this is a functional knock-out, however, how can the response remain to be dose-dependent? What would be the role of TLR2 and TLR4?

In TLR2-/- mice, the contractions are increased upon stimulation with ACh in the presence of  AF-DX 116, and similarly with mecamylamine. What is the meaning of these findings especially given the different background of the mice?

The study claims to identify the signaling and regulation but due to the descriptive nature of the write up falls short of explaining what the signaling cascade would be.

More so, the conclusion drawn from the expression data are not supported by the data presented, especially in the absence of quantification.  

Methods

Western  Blot need to be quantified: M3 is interpreted as being low in TLR2-/- and TLR4-/-, but for both the actin control is much lower than the wild-type- for TLR4-/- the actin control indicates substantially less protein loaded.

To make the statement that β4 expression is only decreased in protein in TLR4-/- not TLR2-/- quantification is necessary.

More so, the conclusions that α3 is diminished in TLR2-/-and TRL4-/- is incorrect: the protein for the TLR4 wild-type control is already low and there is not decrease to be appreciated in the knock-out sample.

Language/Manuscript layout

The abstract lists a number of findings but provides no context to data generated between the muscle contractility studies and expression studies. Is is also unclear why the functional analysis is introduced before the comment about mRNA and protein levels in the manuscript. 

During the entire manuscript, the authors refer to identifying signaling pathways but the hierarchy and impact are lost. A graphical summary would be highly beneficial.

The authors conclude that their study of muscarinic (M2 and M3) and nicotinic (α3, β4, 246 and α7 subunits) ACh receptors mRNA expression in the ileum from WT, TLR2-/- and TLR4-/- mice is the basis for the response induced by Ach in the experimental models. Yet, the muscle contractility in the presence of the antagonists is done prior without any knowledge if the receptors are differentially expressed. Why is there no comparison between the antagonist studies and the receptor expression presented? Can the antagonist work if the receptor is decreased?

Author Response

Reviewer 2

In the manuscript  “TLR2 and TLR4 modulate mouse ileal motility by the regulation of muscarinic and nicotinic receptors”, Layunta et al. aim to demonstrate that TLR2 and TLR4 signaling modulates the motor responses to ACh in the mouse ileum as the deficiency of TLR2 or TLR4 expression affects the expression pattern of muscarinic and nicotinic ACh receptors.

Data interpretation and conclusions:

Overall, the study design and purpose as well as the conclusions need to be clarified and described in more detail. While the authors described the goal of the study to demonstrate that TLR2 regulates muscarinic M2 and M3 and nicotinic α3β4 Ach receptors and TLR4 M3, α3β4 and  α7 Ach receptors, the ‘signaling and regulation’ are not truly defined.

Answer: We appreciate the indication given by reviewer. We show that TLR2 and TLR4 are involved in the expression of nAChR and mAChR and the deficiency of these TLRs affect the ACh motor response in the ileum of mice. We agree that the signaling is not fully defined. In consequence, we have changed this statement in the text for the words “interaction” or “involvement” in the title (lines 2-3) and discussion (lines 409-410, 415-416, 420, 432), since lack of these TLRs affect ACh motor response due to the downregulation of mAChR and nAChR.

If regulation is protein expression, then the data assessing receptors in the two different mouse models need to be moved up to build the foundation for the subsequent study. However, as described below there are improvements required to show convincing evidence that there is a different in receptor expression upon TLR2 and TLR4 knock-out.

Answer: Thank you for the comment. We would like to clarify that our first aim was to investigate if TLR2 and TLR4 are involved in the ileal motor response to ACh. Once we observed that the deficiency of TLR2 and TLR4 induced an alteration (reduction) in the motor response to ACh, our second goal was to address which other signaling molecules could be involved in these effects. That´s why we analyzed the muscarinic and nicotinic ACh receptors, since they are the main receptors involved in the neurotransmission mediated by ACh.

The main line of our research group is the study of intestinal physiology. For this reason, we always approach the study of function first and then molecular studies, as it is shown in the following papers:

Forcen, R.; Latorre, E.; Pardo, J.; Alcalde, A.I.; Murillo, M.D.; Grasa, L. Toll-like receptors 2 and 4 modulate the contractile response induced by serotonin in mouse ileum: analysis of the serotonin receptors involved. Neurogastroenterology and motility : the official journal of the European Gastrointestinal Motility Society 2015, 27, 1258-1266, doi:10.1111/nmo.12619.

Forcen, R.; Latorre, E.; Pardo, J.; Alcalde, A.I.; Murillo, M.D.; Grasa, L. Toll-like receptors 2 and 4 exert opposite effects on the contractile response induced by serotonin in mouse colon: role of serotonin receptors. Experimental physiology 2016, 101, 1064-1074, doi:10.1113/EP085668.

Grasa, L.; Abecia, L.; Pena-Cearra, A.; Robles, S.; Layunta, E.; Latorre, E.; Mesonero, J.E.; Forcen, R. TLR2 and TLR4 interact with sulfide system in the modulation of mouse colonic motility. Neurogastroenterology and motility : the official journal of the European Gastrointestinal Motility Society 2019, 31, e13648, doi:10.1111/nmo.13648.

The authors should address if in the absence of knowledge about receptor presence the response to Ach in Figures 1-2 is meaningful.

Answer: We appreciate the suggestion of reviewer. Statistical analysis are shown in the text of the manuscript (lines 214-217, 224-232, 257-260). However, and for a better understanding of the work, we have added the statistical analysis in Figures 1 and 2.

In terms of signaling, the authors demonstrate a reduced effect of Ach on muscle contractility in both knock-out models compared to wild-type. If this is a functional knock-out, however, how can the response remain to be dose-dependent? What would be the role of TLR2 and TLR4?

Answer: Thanks for the question. These knockout models have a constitutive loss of TLR2 (TLR2-/-) or TLR4 (TLR4-/-) receptors, respectively. As we show in this work, the deletion of TLR2 or TLR4 downregulate the mRNA and protein expression of the muscarinic and nicotinic ACh receptors, but they are still expressed in the ileum of these mice and are functional. This fact would explain the dose-dependent response to ACh, although in a lower degree.

In TLR2-/- mice, the contractions are increased upon stimulation with ACh in the presence of  AF-DX 116, and similarly with mecamylamine. What is the meaning of these findings especially given the different background of the mice?

Answer: Thanks for the question. We agree with the reviewer that the finding showing that the response to ACh is increased in the presence of M2 and α3β4 blockers in TLR2-/- mice, without modifying the expression of mRNA and protein of these receptors, is surprising and represents a limitation to our study. However, this result could be explained by the fact that it has been described that both muscarinic receptors and nicotinic receptors may be subjected to post-translational modifications that could modify the affinity of the receptors for ACh and, therefore, its function. The lack of TLR2 gene in these animals could have induced these post-translational modifications. In fact, post-translational modifications have been described in G protein–coupled receptors (Alfonzo-Mendez et al., 2016) and specifically, an N-glycosylation seems to be key in determining M3 receptor distribution and localization (Romero-Fernandez et al., 2011). On the other hand, three posttranslational modifications are known for the nicotinic acetylcholine receptor family: glycosylation, phosphorylation and palmitoylation (Albuquerque et al., 2009).

Albuquerque EX, Pereira EF, Alkondon M, Rogers SW (2009). Mammalian nicotinic acetylcholine receptors: from structure to function. Physiological reviews 89(1): 73-120.

Alfonzo-Mendez MA, Alcantara-Hernandez R, Garcia-Sainz JA (2016). Novel Structural Approaches to Study GPCR Regulation. International journal of molecular sciences 18(1).

Romero-Fernandez W, Borroto-Escuela DO, Alea MP, Garcia-Mesa Y, Garriga P (2011). Altered trafficking and unfolded protein response induction as a result of M3 muscarinic receptor impaired N-glycosylation. Glycobiology 21(12): 1663-1672.

We have included this information in the Discussion (page 12, lines 412-422) and references (citations 44-46)

The study claims to identify the signaling and regulation but due to the descriptive nature of the write up falls short of explaining what the signaling cascade would be.

Answer: We appreciate the indication given by reviewer. We show that TLR2 and TLR4 are involved in the expression of nAChR and mAChR and the deficiency of these TLRs affect the ACh motor response in the ileum of mice. We agree that the signaling is not fully defined. In consequence, we have changed this statement in the text for the words “interaction” or “involvement” in the title (lines 2-3) and discussion (lines 409-410, 415-416, 420, 432), since lack of these TLRs affects ACh motor response due to the downregulation of mAChR and nAChR.

More so, the conclusion drawn from the expression data are not supported by the data presented, especially in the absence of quantification. 

Answer: Thank you for the comment. We would like to clarify that expression data of mRNA and protein are included in the graphs and western blot images of Figures 3, 4 and 5. The way in which the data have been quantified are described in materials and methods (section 2.3. Gene expression by real-time PCR and section 2.4. Western blotting).

We conclude that the lack of TLR2 and TLR4 affects the ileal motility (Figure 1 and 2, with statistical analysis in the text, and now, also in the graphs) and this is due to the mRNA and protein downregulation of the ACh receptors in ileum of mice (Figures 3, 4 and 5; with statistical analysis included in the graphs). In this way, our conclusion is supported by the data presented and analyzed.

Methods

Western  Blot need to be quantified: M3 is interpreted as being low in TLR2-/- and TLR4-/-, but for both the actin control is much lower than the wild-type- for TLR4-/- the actin control indicates substantially less protein loaded.

Answer: Thank you for the comment. We would like to clarify that M3 protein expression quantification was carried out in at least 5 animals per group by densitometry analysis and arbitrary units (% of control). As it is shown in Figure 5A, the results were 111% in WT, 31.93% in TLR2-/- and 51.83% in TLR4-/-. We agree that apparently, in TLR2-/- and TLR4-/- animals, it seems a lower protein load, but does not explain the huge decrement of M3 signal in both conditions compared with the control. In addition, the statistics indicated a significant difference by using Mann-Whitney U-test.

To make the statement that β4 expression is only decreased in protein in TLR4-/- not TLR2-/- quantification is necessary.

Answer: Thank you for the comment. As we have explained above, the β4 protein expression quantification was carried out in at least 5 animals per group by densitometry analysis and arbitrary units (% of control). As it is shown in Figure 5C, the results were 100.63% in WT, 92.85% in TLR2-/- and 60.48% in TLR4-/-. We agree that maybe is difficult to see visually the downregulation in TLR4-/- animals, but this can be due to the band has a lower number of pixels. The densitometry of the bands have been analyzed with the software Quantity One 1-D following the manufacture instructions and indicating a lower density of β4 protein expression in the TLR4-/- animal compared with the wildtype by using Mann-Whitney U-test.

More so, the conclusions that α3 is diminished in TLR2-/-and TRL4-/- is incorrect: the protein for the TLR4 wild-type control is already low and there is not decrease to be appreciated in the knock-out sample.

Answer: Thank you for the comment. As we have explained above, the α3 protein expression quantification was carried out in at least 5 animals per group by densitometry analysis and arbitrary units (% of control).  As it is shown in Figure 5B, the results were 129.0 % in WT, 46.26 % in TLR2-/- and 55.43 % in TLR4-/-. We agree that maybe is difficult to see visually the downregulation in TLR4-/- animals respect to the second WT in the gel we show, but it can be seen more easily respect to the first WT of the gel.  We want to show in the same gel the results for TLR2-/- and TLR4-/-, without modifying or cutting the gel. The densitometry of the bands have been analyzed with the software Quantity One 1-D following the manufacture instructions and indicating a lower density of β4 protein expression in the TLR4-/- animal compared with the wildtype by using Mann-Whitney U-test.

 Language/Manuscript layout

The abstract lists a number of findings but provides no context to data generated between the muscle contractility studies and expression studies. Is is also unclear why the functional analysis is introduced before the comment about mRNA and protein levels in the manuscript.

Answer: Thank you for the comment. As we explained before, we would like to clarify that our first aim was to investigate if TLR2 and TLR4 are involved in the ileal motor response to ACh. Once we observed that the deficiency of TLR2 and TLR4 induced an alteration (reduction) in the motor response to ACh, our second goal was to address which other signaling molecules could be involved in these effects. That´s why we analyzed the muscarinic and nicotinic ACh receptors, since they are the main receptors involved in the neurotransmission mediated by ACh. The main line of our research group is the study of intestinal physiology. For this reason, we always approach the study of function first and then molecular studies.

During the entire manuscript, the authors refer to identifying signaling pathways but the hierarchy and impact are lost. A graphical summary would be highly beneficial.

Answer: We appreciate the suggestion given by the reviewer. As we have explained before, we agree that the signaling is not fully defined. In consequence, we have changed this statement in the text for the words “interaction” or “involvement” in the title (lines 2-3) and discussion (lines 409-410, 415-416, 420, 432), since lack of these TLRs affect ACh motor response due to the downregulation of mAChR and nAChR.

As the reviewer suggests, we have included a graphical abstract showing the context and findings of this work.

The authors conclude that their study of muscarinic (M2 and M3) and nicotinic (α3, β4, 246 and α7 subunits) ACh receptors mRNA expression in the ileum from WT, TLR2-/- and TLR4-/- mice is the basis for the response induced by Ach in the experimental models. Yet, the muscle contractility in the presence of the antagonists is done prior without any knowledge if the receptors are differentially expressed. Why is there no comparison between the antagonist studies and the receptor expression presented? Can the antagonist work if the receptor is decreased?

Answer: We appreciate these comments

“The muscle contractility in the presence of the antagonists is done prior without any knowledge if the receptors are differentially expressed.”

As we explained before, we would like to clarify that our first aim was to investigate if TLR2 and TLR4 are involved in the ileal motor response to ACh. Once we observed that the deficiency of TLR2 and TLR4 induced an alteration (reduction) in the motor response to ACh, our second goal was to address which other signaling molecules could be involved in these effects. That´s why we analyzed the muscarinic and nicotinic ACh receptors, since they are the main receptors involved in the neurotransmission mediated by ACh. The main line of our research group is the study of intestinal physiology. For this reason, we always approach the study of function first and then molecular studies.

“Why is there no comparison between the antagonist studies and the receptor expression presented? Can the antagonist work if the receptor is decreased?”

We would like to clarify that the motility experiments have been performed in vitro in an organ bath. Acetylcholine and its antagonists were added to the bath and the effect of these drugs on intestinal smooth muscle was studied. Each ileum segment spent at least 3 hours in the organ bath and was therefore subjected to a situation of stress and tissue damage. For this reason, we believe that the mRNA levels under these conditions would be altered by environmental conditions and would not correspond to reality.

Round 2

Reviewer 2 Report

This revised version is improved.